# Genome-Wide Analysis of Alternative Splicing Events Responding to High Temperatures in *Populus tomentosa* Carr.

Xue Wang [1,†], Yan Wang [2,†], Ruixue Wang [2], Longfeng Gong [1], Lei Wang [1] and Jichen Xu [1,2,*]

1   State Key Laboratory of Tree Genetics and Breeding, College of Biological Sciences and Technology, Beijing Forestry University, Beijing 100083, China
2   National Engineering Research Center of Tree Breeding and Ecological Restoration, College of Biological Sciences and Technology, Beijing Forestry University, Beijing 100083, China
*   Correspondence: jcxu282@sina.com; Tel.: +86-10-6233-6628
†   These authors contributed equally to this work.

**Abstract:** Through alternative splicing (AS) processes, eukaryotic genes can generate a variety of transcription isoforms that lower the expression levels of the normal transcripts or result in diversity in the genes' activities. Then, AS plays a significant role in the control of plant development and stress tolerance. In this study, we analyzed *Populus tomentosa* Carr. TC1521's AS episodes in response to high temperatures. The samples treated at 25 °C, 30 °C, 35 °C, and 40 °C produced a total of 10,418, 11,202, 9947, and 14,121 AS events, respectively, which responded to 4105, 4276, 4079, and 4915 genes, respectively, representing 9.84%, 10.25%, 9.78%, and 11.78% of the total number of transcribed genes, respectively. The most common AS pattern, accounting for 42.31% to 51.00% of all AS events, was intron retention (IR), followed by exon skipping (ES), which accounted for 9.14% to 10.23% of all AS events. respectively. According to sequence characterization, AS was negatively correlated with guanine-cytosine content (GC content) but favorably correlated with intron length, exon number, exon length, and gene transcription level. Compared to treatment at 25 °C, 2001 distinct AS genes were discovered at 40 °C. They were primarily enriched in the RNA degradation pathway and the valine, leucine, and isoleucine degradation route, according to (gene ontology) GO and Kyoto Encyclopedia of Genes and Genomes (KEGG) enrichment analysis. These findings demonstrated how the AS process might be severely impacted by high temperatures. In addition, the information on AS isoforms helped us comprehend stress-resistance mechanisms in new ways and completed molecular design breeding.

**Keywords:** *Populus tomentosa* Carr.; alternative splicing; high temperature; RNA degradation; isoform

## 1. Introduction

In plant genomes, most genes contain intron sequences and have conserved intron patterns. (The bases at both sides are generally 5′GT-AG3′.) [1]. The initial product of gene transcription is the precursor mRNA (pre-mRNA), in which intron fragments are further removed through a splicing protein complex and the exons are religated to form mature protein-coding mRNA. Therefore, accurate splicing of the pre-mRNA is essential for the integrity and function of proteins and necessary for the plants' growth and development and their adaptation to the environment.

According to numerous studies, the pre-mRNA also experiences a variety of alternative-splicing (AS) events, producing various mature mRNA products [2,3]. *OsHSFA2dI* and *OsHSFA2dII*, for instance, are two splicing isoforms of the rice heat-shock transcription factor *OsHSFA2d*. *OsHSFA2dII* is the dominant but transcriptionally inactive splicing isoform under typical circumstances. *OsHSFA2d* is alternatively spliced into the transcriptionally active isoform *OsHSFA2dI*, which takes part in a plant's response to heat stress [4]. When lacking DNA-binding domains, a myelobastosis (MYB) transcription factor *MaMYB16* in banana fruit experienced AS and produced the alternative transcript *MaMYB16S* in

addition to the usual transcript *MaMYB16L*. *MaMYB16L* acts as a transcription suppressor that may inhibit the expression of genes related to starch degradation and transcription factor *MaDREB2* [5]. *MaMBY16S* can compete with *MaMYB16L* to bind and form inactive heterodimers, reducing the transcriptional inhibition of *MaMYB16L* on the target genes in fruit ripening. Hence, AS is important for the post-transcriptional regulation of the genes or for the diversification of the functions of the genes [6–8].

AS is heavily involved in all phases of plant development. In an examination of 28 soybean samples taken at different developmental phases using RNA-sequencing technology, more than 63% of genes with multiple exons underwent an AS procedure, [9]. An analysis of bamboo shoot transcripts revealed that 60.74% of the genes were affected by AS events, with winter bamboo samples undergoing more AS processes than samples from early, late, and mature development periods. The fact that 26 of the samples had serine/arginine-rich genes that produced 109 isoforms throughout the four periods showed that AS was involved in the growth and development of the bamboo shoot [10].

In the xylems of poplar and eucalyptus, during wood formation, 25% (poplar) and 26.8% (eucalyptus) of the AS events changed the protein domain, and 28.3% (poplar) and 20.7% (eucalyptus) of the highly expressed transcripts were spliced alternatively, changing the original reading frame. Seventy-one AS events, meanwhile, were retained in both species [11].

In addition, AS is a reaction to stress load. After salt-stress treatment, it was discovered that 11141 genes in wheat had significant AS alterations, indicating that AS control may be crucial for salt-stress defense [12]. In *Cassava*, 3292 and 1025 AS occurrences in cold and drought stress settings, respectively, were observed. Premature termination codons were found in 58.5% of the AS transcripts under cold stress, demonstrating the intricacy and specificity of gene regulation in abiotic stress [13]. More than 48,000 novel transcripts were particularly expressed in maize under drought conditions when leaves experienced AS events at a higher rate than that of corncobs [14].

Therefore, plant tissue types, developmental stages, and stress conditions all play a significant role in AS occurrences. For plants to control their growth and withstand stress, AS is extremely important [15–17].

The woody model plant, poplar, has excellent environmental tolerance. This study aims to investigate how high temperatures affect the AS of *Populus tomentosa* genes based on RNA-seq. The knowledge of the AS isoforms of genes will enhance our comprehension of the molecular mechanisms that protect plants from heat stress and provide us with access to fresher gene resources for molecular breeding programs.

## 2. Materials and Methods

### 2.1. Plant Material and Treatment

*Populus tomentosa* TC1521 clonal seedlings were planted in pots (20 × 20 cm) in an artificial climate incubator, with a 1:1 mixture of vermiculite and vegetative soil as the culture substrate. The growth conditions were 25 °C and a 16 h light/8 h dark photoperiod with a light intensity of 150 $\mu$mol m$^{-2}$ s$^{-1}$. The plants were irrigated with water every 3 days and irrigated with Hoagland nutrient solution once a week [18].

### 2.2. Physiological-Index Test

The poplar plants with heights of 50 cm were treated at 25 °C, 30 °C, 35 °C, 40 °C for 24 h. The mature poplar leaves were sampled and used for physiological-index measurements. Three biological replicates were set for each treatment. Significantly different performances of the control plants (25 °C) and heat-treatment plants were determined by SPSS24.0 software at $p$ value of 0.01.

Fresh leaves (0.1 g) were cut into small pieces (0.5 cm × 0.5 cm) and soaked in 30 mL deionized water at 25 °C and shaken at 180 rpm for 1 d. The conductivity was measured as R1 by the electrical conductivity meter (DDS-11C). The samples were then autoclaved

at 121 °C for 20 min and shaken for 24 h. The conductivity was measured as R2. Relative electrical leakage (REL) was calculated by R1/R2.

Fresh leaves (0.1 g) were ground in 1 mL 10% trichloroacetic acid. The homogenization was centrifuged at 12,000 rpm for 5 min. The supernatant was collected and mixed with 1 mL 0.6% thiobarbituric acid, kept in boiling water bath for 15 min, and centrifuged at 12,000 rpm for 5 min. The absorbance of supernatant was measured at 450, 532, and 600 nm. The malondialdehyde content (MDA) content was calculated according to the following formula: $[6.45 \times (OD_{532} - OD_{600}) - 0.56 \times OD_{450}] \times$ total extract volume/fresh weight of sample.

Fresh leaves (0.1 g) were placed in 10 mL dimethyl sulfoxide and enclosed at 25 °C in the dark for 2 days. The light absorption values of the extract solution at wavelengths of 663 nm and 645 nm were determined by an enzyme-labeled instrument, and the chlorophyll content (Chl) was calculated according to the following formula: $[(0.0127 \times A663 - 0.0629 \times A645) + (0.0229 \times A645 - 0.0468 \times A663)] \times$ total extract volume/fresh weight of sample.

The glucose standard solution was prepared at concentrations of 0 μg/mL, 50 μg/mL, 100 μg/mL, and 200 μg/mL. Five mL of anthranone was added to 1 mL standard solutions that were kept in a boiling water bath for 10 min. The absorption values of the standard solutions were measured at a wavelength of 620 nm for a standard curve. The fresh leaves (0.1 g) were sampled and ground, kept in a boiling water bath for 25 min, and centrifuged at 5000 rpm for 10 min. A 50 μL supernatant was diluted to 10 times and mixed with 2.5 mL of anthranone and kept in a boiling water bath for 10 min. The absorbance value was measured at a wavelength of 620 nm. The soluble sugar content (SSC) was determined by comparison with the standard curve.

### 2.3. RNA Extraction, cDNA Library Preparation, and RNA-seq

The total RNA was extracted from the poplar leaves under 24h temperature treatment using a TaKaRa MiniBEST Universal RNA Extraction Kit (Takara Bio Inc, Kusatsu, Japan). RNA quality was evaluated with a NanoDrop2000 spectrophotometer (Thermo Scientific, Waltham, MA, USA). RNA integrity was assessed using the RNA Nano 6000 Assay Kit of the Agilent Bioanalyzer 2100 system (Agilent Tech, Santa Clara, CA, USA). An RNA sequencing library was constructed using a NEBNext UltraTM RNA Library Prep Kit (Ipswich, MA, USA). The library was sequenced by the Biomaker company, Taizhou, China. The clustering of the index-coded samples was performed on a cBot Cluster Generation System using a TruSeq PE Cluster Kit v4-cBot-HS (Illumina novaseq6000, PE150). The sequencing was processed on an Illumina platform. The clean reads were obtained from the ploy-N reads by removing the adapter and the low quality reads. Each poplar sample had three biological replicates. Stata software 17.0 was used for statistics and Pearson correlation analysis.

### 2.4. Identification of AS Events

The clean sequencing reads were aligned to the *Populus trichocarpa* genome (v3.1) to define the gene-reads locus (https://phytozome.jgi.doe.gov/pz/portal.html#!info?alias=Org_Ptrichocarpa_er, accessed on 31 November 2018), using TopHat 2.0.10 software. Cufflinks 0.16 software was used to predict transcript-isoforms models. ASprofile 1.0 software was used to classify and count the AS events of each sample. Fragments-per-kilobase of transcript-per-million-fragments mapped (FPKM) were calculated by StringTie 2.1.4 software to display the expression levels of the transcripts or genes. Multiple comparisons of the gene characters of the high-AS genes, low-AS genes, and non-AS genes were carried out via SPSS23.0 software. The significant difference level was at $p = 0.05$.

### 2.5. Functional Annotation and Enrichment Pathway of AS Genes

AS genes in the poplar leaves were aligned with the DAVID database. GOseq software (version 1.0) was used to evaluate the difference between genes and background in each gene ontology (GO) term. The gene enrichment analysis was conducted and defined by

a *p* value < 0.05. The pathway analysis of the AS genes was performed according to the Kyoto Encyclopedia of Genes and Genomes (KEGG) database. The pathways of $p < 0.05$ were defined as significantly enriched.

## 3. Results

### 3.1. P. tomentosa Displayed a Strong Resistance to High Temperature

The poplar tree demonstrated a consistent phenotypic and physiological function at temperatures of 25 °C, 30 °C, and 35 °C. However, the chlorophyll content dramatically reduced by 14.19% at 40 °C compared to the chlorophyll content at 25 °C, while the amounts of soluble sugar, MDA content, and relative electrical leakage increased by 22.96%, 14.03%, and 27.16%, respectively (Figure 1). This suggests that *P. tomentosa* TC1521 has a critical temperature range of 40 °C.

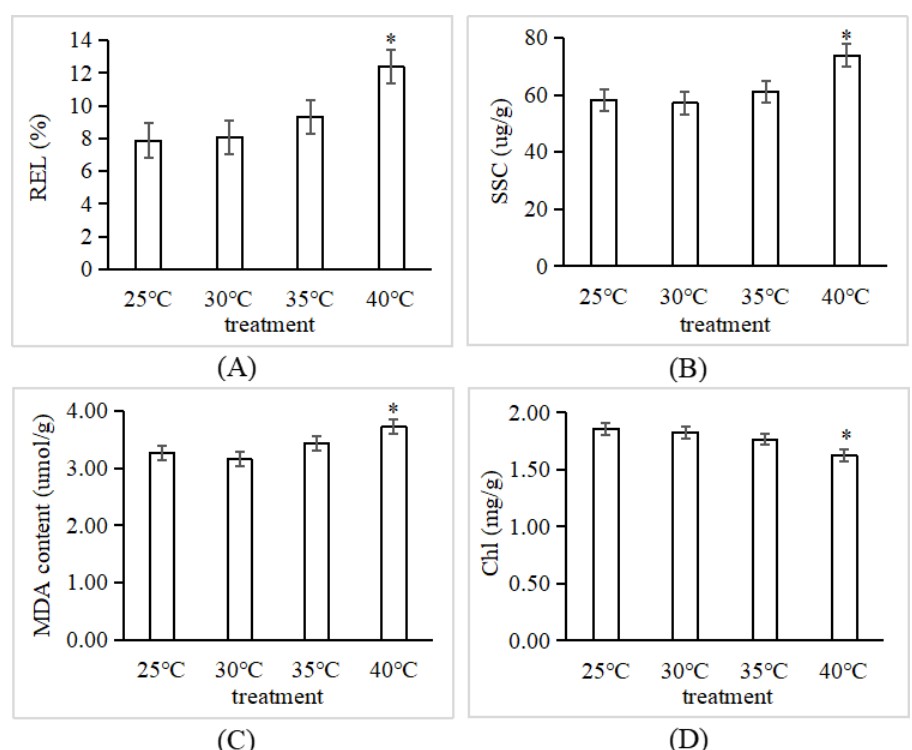

**Figure 1.** Physiological performance of *P. tomentosa* under heat treatments. * indicate a significant difference between the treatments (30 °C, 35 °C, 40 °C) and control (25 °C) ($p < 0.05$, and $p < 0.01$), respectively. (**A**), relative electrical leakage (REL); (**B**), soluble sugar content (SSC); (**C**), malondialdehyde (MDA) content; (**D**), chlorophyll content (Chl).

### 3.2. Overview of RNA-seq Data

By using RNA-seq, 12 samples of poplar leaves yielded a total of 89.63 Gb in clean data, an average of 7.47 Gb per sample. The sequencing data were of high quality, as the percentage of Q30 bases was higher than 94.98%. The transcripts' average GC content was 44.72%. A total of 73.66% of the clean reads were mapped to the *P. trichocarpa* reference genome, and 70.72% of those reads were assigned to a single locus, with the remaining 2.94% being mapped to several loci. mRNAs made up the majority of the unique mapped reads, along with tRNAs and ncRNAs. The majority of the repeat-associated RNAs, tRNAs, and rRNAs were found in the multiple mapped reads (Table 1).

**Table 1.** Statistics of sequence and alignment with the reference genome.

| Items | 25 °C | 30 °C | 35 °C | 40 °C |
|---|---|---|---|---|
| Clean reads (bp) | 22,466,189 | 22,478,489 | 22,692,788 | 21,995,836 |
| GC content | 44.94% | 44.96% | 45.22% | 43.78% |
| % ≥ Q30 | 94.98% | 95.04% | 95.36% | 95.20% |
| Mapped reads (bp) | 16,741,731 | 16,669,136 | 16,990,918 | 15,627,313 |
| Unique mapped reads (bp) | 16,071,329 | 15,956,074 | 16,233,291 | 15,132,363 |
| Multiple mapped reads (bp) | 670,403 | 713,063 | 757,628 | 494,950 |

Note: % ≥ Q30, proportion of bases with a quality value of 30 or greater in clean data; mapped reads, the number of reads mapped to the genome; unique mapped reads, the number of reads mapped to a unique location in the genome; multiple mapped reads, the number of reads mapped to multiple locations in the genome.

### 3.3. AS Events in Response to High Temperatures in Poplar

*P. tomentosa* transcripts were compared to the reference genome, and 10,418, 11,202, 9947, and 14,121 AS events responding to 4105, 4276, 4079, and 4915 genes were discovered in the leaf samples treated at 25 °C, 30 °C, 35 °C, and 40 °C, respectively. These AS genes made up 9.84%, 10.25%, 9.78%, and 11.78%, respectively, of the total number of transcribed genes. *P. tomentosa* TC1521 experienced more AS episodes due to the severe temperature situation (40 °C).

Exon skipping (ES), intron retention (IR), alternative 5′ splicing site (A5SS), and alternative 3′ splicing site (A3SS) were used to categorize all AS events found in poplar samples (Figure S1). Of the AS events, 42.31%–51.00% were of the IR kind. The least common event type, the ES type, made up 9.14%−10.22% of all AS events (Table 2). Under various temperature treatments, the samples' AS type modes remained constant.

**Table 2.** Distribution of the alternative splicing events in *P. tomentosa* TC1521 at different temperatures.

| Treatment Temperature | AS Events | IR | A5SS | A3SS | ES | Sum |
|---|---|---|---|---|---|---|
| 25 °C | AS events | 4659 | 2723 | 2042 | 994 | 10,418 |
| | Ratio (%) * | 44.72 | 26.14 | 19.6 | 9.54 | |
| | AS genes | 2599 | 1271 | 965 | 759 | 4105 |
| 30 °C | AS events | 5713 | 2531 | 1931 | 1027 | 11,202 |
| | Ratio (%) * | 51.00 | 22.59 | 17.24 | 9.17 | |
| | AS genes | 2723 | 1323 | 1076 | 792 | 4276 |
| 35 °C | AS events | 4377 | 2760 | 1901 | 909 | 9947 |
| | Ratio (%) * | 44.00 | 27.75 | 19.11 | 9.14 | |
| | AS genes | 2550 | 1231 | 994 | 708 | 4079 |
| 40 °C | AS events | 5975 | 3735 | 2968 | 1443 | 14,121 |
| | Ratio (%) * | 42.31 | 26.45 | 21.02 | 10.22 | |
| | AS genes | 3055 | 1743 | 1469 | 1078 | 4915 |

* The values were obtained by the formula of the relevant type of AS events divided by the total AS events.

### 3.4. Structural Analysis of AS Genes in Poplars

All transcribed genes were divided into three groups based on the number of AS events for each gene: high-AS genes (AS events ≥ 5), low-AS genes (4 ≥ AS events ≥ 1), and non-AS genes (AS events = 0), totaling 917 genes, 7916 genes, and 31,873 genes, respectively. The exon lengths of the high-AS genes and the low-AS genes were, respectively, 1.92 times and 1.80 times longer than those of the non-AS genes, while the intron lengths of the high-AS genes and low-AS genes were, respectively, 3.08 times and 2.65 times longer than those of the non-AS genes. The average expression levels of the high-AS genes and the low-AS genes were 1.46 times and 1.26 times higher, respectively, than the expression level of the non-AS genes, and the numbers of exons in the high-AS genes and the low-AS genes were 2.85 times and 1.96 times greater, respectively, than the number of the non-AS genes. The average guanine-cytosine (GC) contents of the high- and low-AS genes, however, were

0.947 times and 0.950 times the average GC content of the non-AS genes. For instance, the high-AS gene *Potri.002G032900.v3.0* had six transcript isoforms. It had seven exons, with an average exon length of 1362 bp, an average intron length of 3190 bp, a GC content of 37.81%, and an FPKM value of 327.34. *Potri.007G021300.v3.0*, in contrast, was a low-AS gene with two transcript isoforms. It had eight exons, with an average exon length of 1095 bp and an average intron length of 1206 bp. It also had a GC content of 38.07% and a total FPKM value of 58.84. *Potri.008G121700.v3.0* was a non-AS gene and had three exons with an average exon length of 999 bp, an intron length of 806 bp, a GC content of 40.66%, and a total FPKM value of only 16.04. Obviously, the frequency of AS events was in positive association with the intron length, the exon length, the exon number, and the transcript level, but in negative association with GC content (Figure 2).

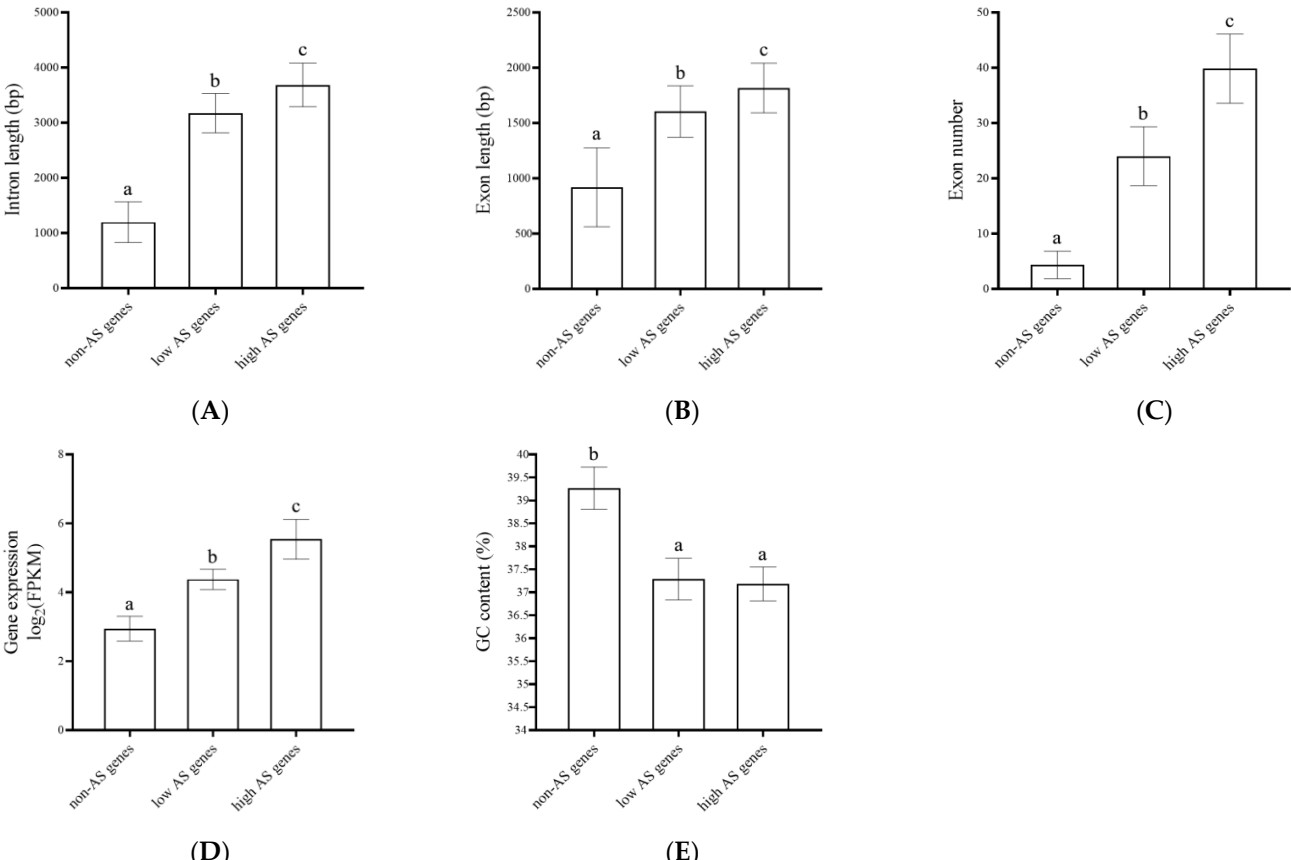

**Figure 2.** Multiple comparison of the gene characters among the high-AS genes, the low-AS genes, and the non-AS genes. (**A**) intron length; (**B**) exon length; (**C**) exon number; (**D**) gene expression level; (**E**) GC content. The different letters on the columns indicate their significant difference levels ($p < 0.05$).

### 3.5. AS Events Induced by Heat Stress and Function Annotation

The treatment of 30 °C, 35 °C, and 40 °C produced 5688, 5067, and 8767 specific AS events, respectively, compared to the AS events in the control temperature of 25 °C, responding to 1403, 1303, and 2001 AS genes (Figure 3): 3320 AS events occurred in all four temperature conditions, 804 AS events occurred in three of the circumstances (35 °C, 40 °C, and 30 °C), 916 AS events occurred in two of the situations (35 °C, 40 °C), and 6066 AS events occurred only in the condition of 40 °C. Higher temperatures caused a considerable increase in AS occurrences in poplar trees.

The 2001 genes were mapped to gene ontology (GO) concepts in response to the 8767 specific AS events brought on by 40 °C stress (Figure 4). Twenty biological processes, including cellular processes, metabolic processes, and single-organism processes, were

applied to 1033 genes. In 15 categories of cellular components, including cell portions, cells, organelles, and membranes, 588 genes were functional. In addition, 14 molecular functions, including binding and catalysis, were involved in 415 genes.

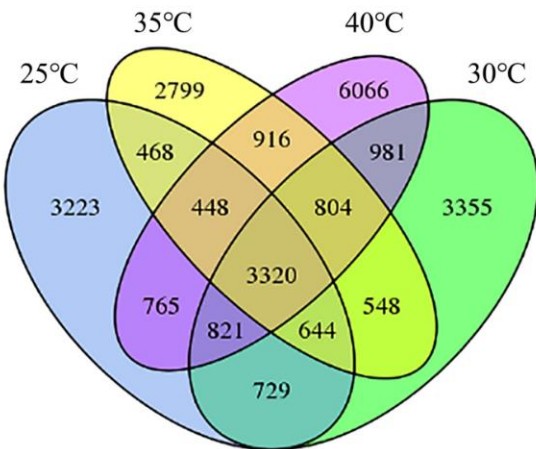

**Figure 3.** Comparison of the differential AS events in poplar leaves at different temperatures.

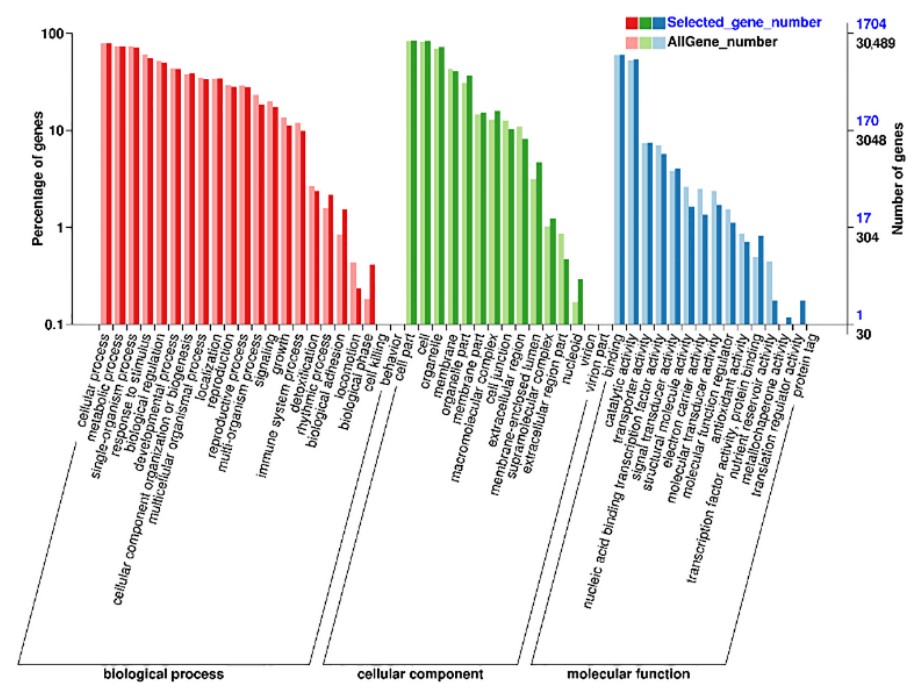

**Figure 4.** 2001 specific AS gene ontologies under 40 °C stress, compared to control under 25 °C stress.

KEGG pathways received assignments for 470 of the 2001 AS genes (Figure 5). Among them, the pathways for the breakdown of RNA and the amino acids valine, leucine, and isoleucine were considerably enriched. In the RNA degradation pathway, which was crucial for eliminating non-functional RNAs and reducing their interference with homologous transcripts, 24 AS genes participated (Figure 6). The eukaryotic core exosome, the exosome coactivator complexes, the cytoplasmic deadenylation, the ski complex, the decapping complex, the lsm complex, and the 5′ exonuclease were primarily structured by them. These AS occurrences also occurred under other hot situations and grew considerably as the temperatures rose. For instance, the *RRP6L2* gene experienced 1, 2, 3, and 6 AS events at 25 °C, 30 °C, 35 °C, and 40 °C, respectively (Table 3). At 25 °C, 30 °C, 35 °C, and 40 °C, on average, 1.29, 1.67, 1.63, and 2.46 AS events occurred, respectively, for all 24 genes. Additionally, the splicing model tended to diversify as the temperatures rose. For example,

the *RRP6L2* gene only exhibited one form of ES at 25 °C, but displayed two types of ES plus IR at 40 °C (Figure 7). The high-temperature-induced AS events that decreased the levels of normal transcripts and waste RNAs in the cells may have contributed to the decrease in plant vigor.

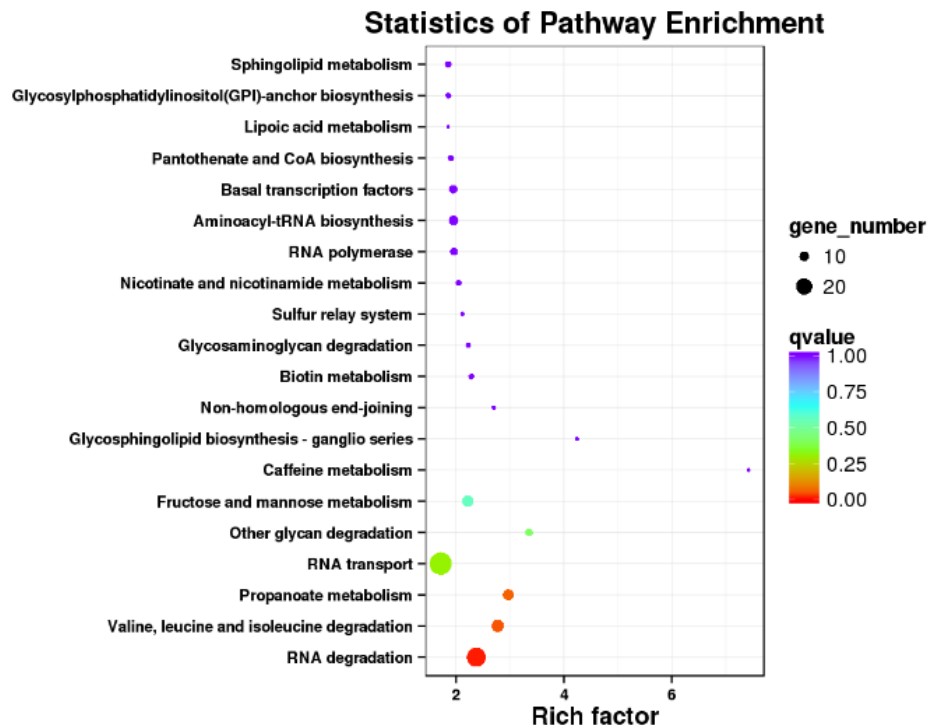

**Figure 5.** Enriched KEGG pathways of 2001 specific AS genes under 40 °C stress, compared to control at 25 °C stress.

**Table 3.** The number of AS events of genes in the RNA degradation pathway at different temperatures.

| Gene | 25 °C | 30 °C | 35 °C | 40 °C |
|------|-------|-------|-------|-------|
| *RRP40* | 1 | 1 | 1 | 2 |
| *CHLH* | 1 | 1 | 1 | 1 |
| *RRP45B* | 2 | 4 | 4 | 4 |
| *RRP44A* | 2 | 2 | 1 | 3 |
| *RRP6L2* | 1 | 2 | 3 | 6 |
| *DCP2* | 1 | 1 | 1 | 1 |
| *XRN3* | 2 | 3 | 4 | 3 |
| *URT1* | 1 | 2 | 1 | 2 |
| *SKI2* | 2 | 4 | 3 | 8 |
| *HVT1* | 1 | 1 | 1 | 1 |
| *VIP2* | 1 | 1 | 2 | 2 |
| *ABCI19* | 2 | 2 | 1 | 1 |
| *ABCI20* | 2 | 1 | 1 | 3 |
| *LSM1B* | 1 | 1 | 2 | 2 |
| *LSM4* | 1 | 1 | 2 | 2 |
| *PFK2* | 1 | 1 | 1 | 1 |
| *PFK3* | 1 | 2 | 1 | 2 |
| *PFK4* | 2 | 2 | 2 | 4 |
| *PFK5* | 1 | 1 | 1 | 2 |
| *CPN60B2* | 1 | 1 | 1 | 1 |
| *CPN60-2* | 1 | 1 | 1 | 1 |
| *Potri.019G032800* | 1 | 1 | 2 | 2 |
| *Potri.009G033400* | 1 | 3 | 1 | 3 |
| *Potri.006G208600* | 1 | 1 | 1 | 2 |

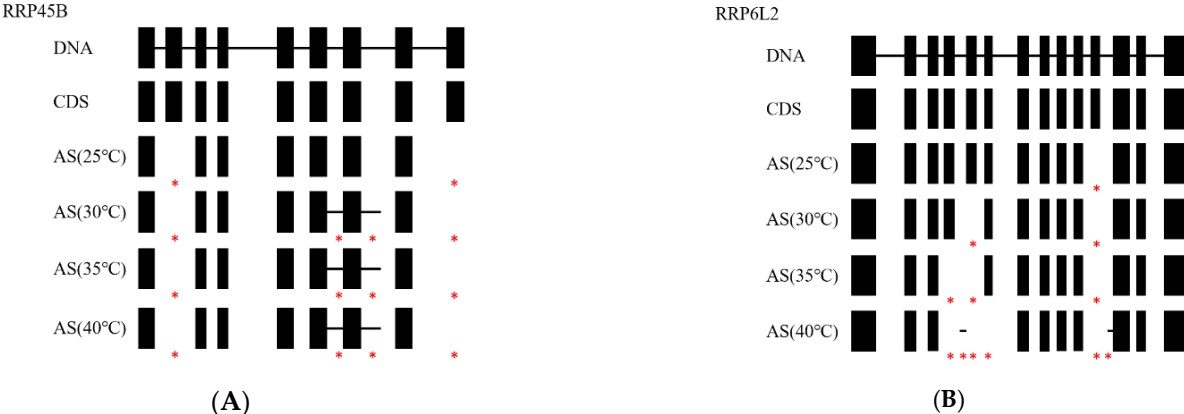

**Figure 6.** RNA degradation pathway and related AS gene distribution (purple) under 40 °C stress. The purple color indicates the AS genes in the pathway.

**Figure 7.** *Cont.*

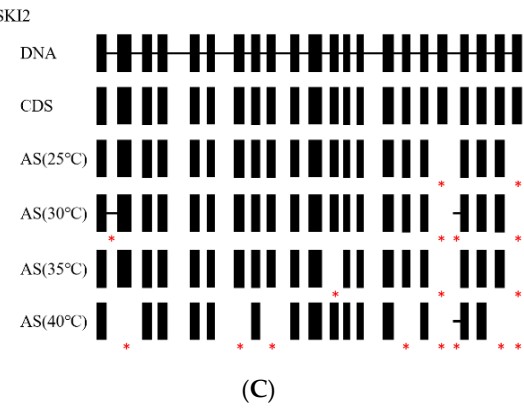
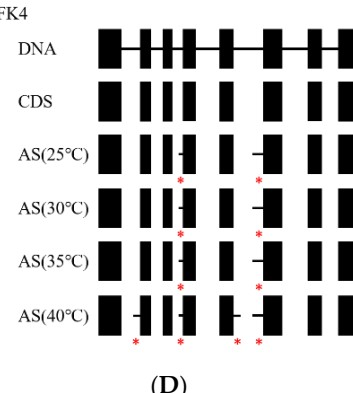

**Figure 7.** AS events of the genes at different temperatures, compared to the DNA sequence and normal coding sequencing (CDS). The blocks represent exons, and the lines between the blocks represent introns. * means a AS event occurred under each temperature condition. (**A**), *RRP45B* gene; (**B**), *RRP6L2* gene; (**C**), *SKI2* gene; (**D**), *PFK4* gene.

## 4. Discussion

### 4.1. AS Events Were Specific to the Environmental Conditions

Abiotic stress has the potential to drastically activate gene AS events. As an illustration, 14,207 AS events were found in *Arabidopsis thaliana* seedlings treated with salt, compared to 12,236 AS events in the control, an increase of 16.11%, with the greatest increase in the ES events [19]. Tea plants experienced increases in AS events of 30,212, 30,325, and 30,552 at different low temperature treatments, while only 27,234 AS events were experienced under normal conditions. IR events experienced the greatest increases, increasing by 9.86%, 10.19%, and 12.18%, respectively [20]. In a wild salinity-tolerant cotton species (*Gossypium davidsonii*) under salt treatment, 12,379 AS events were revealed. while 11,889 AS events were present in control and 4108 AS events were induced by salt, in a ratio of 33.19%; A5SS events showed the greatest increase [21]. In wheat, responses to heat stress, drought stress, and their combination were detected as 251, 6681, and 7451 AS events, respectively [22]. Under cold and drought stress, 3292 and 1025 AS events in cassava, respectively, were significantly triggered [13]. In this investigation, we identified 10418 AS events at 25 °C and 14,121 AS events at 40 °C in poplar, with an increase of 35.54% by high temperature; A3SS events showed the greatest increase. As a result, there was a strong correlation between species and stress levels and the frequency and kind of AS of genes. Next, we considered the importance of further investigating the relevant mechanisms.

### 4.2. AS Events in the RNA Degradation Pathway Particularly Affect the Plant Performance in Abiotic Stress

Stress-related AS events were prevalent, which suggested that more undesirable or faulty RNAs were generated [23]. The effective breakdown of these waste products is essential for maintaining cellular homeostasis under stress [24,25]. The most common method of mRNA degradation in eukaryotic cells is deadenylation, followed by decapitation and destruction, either in the 5′ to 3′ direction or in the 3′ to 5′ direction by exoribonuclease. Some genes related to plant development and stress tolerance have been described as engaged in these processes [26]. For instance, the RNA degradation pathway of *Capsicum annuum* requires a combination of CCR4 and CAF1 [27]. *CaCAF1* overexpression may greatly accelerate tomato plant growth and thicken leaves [28]. At the seedling cotyledon stage in *Arabidopsis*, the null mutants of *DCP1, DCP2*, and *VCS* exhibit a similar lethal phenotype, suggesting that postembryonic development may depend on the mRNA turnover mediated by the decapping complex [29]. RNase III-mediated decay (RMD), which is overactivated by salt stress, is a further post-transcriptional regulatory mechanism that affects *BDF2* [30]. Therefore, RNA degradation-related genes' AS events may have a significant

impact on how well plants perform under stress [31]. According to several studies, the AS genes were considerably enriched in the RNA degradation pathway, such as that in tea plants under high temperatures and drought stress [32]. Similarly, in our work, a total of 24 RNA degradation-related genes experienced AS, and they also displayed a diversity of AS patterns as the temperatures rose. The 24 genes had 2.46 AS events on average at 40 °C, compared to 1.29, 1.67, and 1.63 AS events, respectively, at 25 °C, 30 °C, and 35 °C.

The increased AS events in the RNA degradation pathway at 40 °C indicated that there were more waste products in cells that would be seriously harmful to the regular metabolic activity. This result was consistent with the physiological responses of poplar plants to heat stress, which showed that AS events in the RNA degradation pathway was noteworthy in abiotic resistance. This most likely provides us with a fresh perspective on how stress resistance works and also provides us with some ideas for a molecular breeding program. Future research into the AS patterns, expression, regulation, and responses of the genes in the RNA degradation pathway will be conducted.

## 5. Conclusions

With increases in external temperature, more alternative splicing (AS) events and more complex AS patterns were present in the transcripts of *Populus tomentosa*, indicating that AS is quite involved in the regulation of plant resistance to abiotic stress. These AS genes were primarily enriched in the RNA degradation pathway and the valine, leucine and isoleucine degradation route, which potentially supports some new gene resources in the molecular breeding project. In addition, AS events occurred more frequently in those genes with low GC content, large intron and exon length, more exons, and high gene-transcription levels. Therefore, we determined the point for stress resistance exploration in plants via AS control.

**Supplementary Materials:** The following supporting information can be downloaded at: https://www.mdpi.com/article/10.3390/f14091878/s1, Figure S1: The representation of different alternative splicing event.

**Author Contributions:** Conceptualization, J.X.; methodology, X.W.; software, L.G. and L.W.; investigation, X.W., Y.W. and R.W.; writing—original draft preparation, X.W. and Y.W.; writing—review and editing, J.X.; supervision, J.X.; funding acquisition, J.X. All authors have read and agreed to the published version of the manuscript.

**Funding:** This research was funded by the National Natural Science Foundation of China (31870648).

**Data Availability Statement:** Publicly available datasets were analyzed in this study. This data can be found in GenBank (accession number: SRR12280784-SRR12280777).

**Acknowledgments:** We thank Lesley Benyon for editing the English text of a draft of this manuscript.

**Conflicts of Interest:** The authors declare no conflict of interest.

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
