# Peer review of "Genome-Wide Analysis of Alternative Splicing Events Responding to High Temperatures in Populus tomentosa Carr."

_forests, doi:10.3390/f14091878_

Round 1
Reviewer 1 Report
The authors predicted alternative splicing events in Populus tomentosa using RNA-seq data. Even though authors performed systematic analysis, I have concerns about the methodology they used.
· Which reference genome they used? Is it from Ensemble? how good the reference genome? Because the whole work is based on this mapping of your reads to the reference genome. Authors should pay more attention for choosing the reference genome.
· I saw the authors used ASprofile, tool published in 2013 which is quite old. Why can’t authors try the new tool developed for predicting splicing event, For reference: https://bioconductor.org/packages/release/bioc/vignettes/ASpli/inst/doc/ASpli.pdf. I strongly suggest the authors to perform splicing event with ASpli tool as well and compare the results.
· I also notice that authors didn’t mention about the alternative promoter usage, which is the key to define isoform of a gene. Please explain.
· Authors should consider adding a supplementary figure about the representation of different alternative splicing event with examples. I suggest a tool name “SpliceV” is easy to work on and give good resolution figures.
· For analyzing gene ontology and pathway, authors can use different tools and compare the results.
Author Response
Reviewer 1
The authors predicted alternative splicing events in Populus tomentosa using RNA-seq data. Even though authors performed systematic analysis, I have concerns about the methodology they used.
Which reference genome they used? Is it from Ensemble? how good the reference genome? Because the whole work is based on this mapping of your reads to the reference genome. Authors should pay more attention for choosing the reference genome.
Reply: We used Populus trichocarpa v3.1 as the reference genome. It was modified in the methods section. The genome is well assembled and widely used for the gene annotation.
I saw the authors used ASprofile tool published in 2013 which is quite old. Why can’t authors try the new tool developed for predicting splicing event, For reference: https://bioconductor.org/packages/release/bioc/vignettes/ASpli/inst/doc/ASpli.pdf. I strongly suggest the authors to perform splicing event with ASpli tool as well and compare the results.
Reply: Thanks for the reviewer’s suggestion. We tried it and found the results similar. ASprofile, used in our manuscript, is also a good tool for AS analysis and widely used up to date (Du et al. Transcriptomic and Functional Analyses Reveal the Different Roles of Vitamins C, E, and K in Regulating Viral Infections in Maize. Int. J. Mol. Sci. 2023, 24, 8012)( Wang et al. The Pitaya Flower Tissue’s Gene Differential Expression Analysis between Self-Incompatible and Self-Compatible Varieties for the Identification of Genes Involved in Self-Incompatibility Regulation. Int. J. Mol. Sci. 2023, 24, 11406.). Therefore, we keep the use here.
I also notice that authors didn’t mention about the alternative promoter usage, which is the key to define isoform of a gene. Please explain.
Reply: In the part “ 3.4. Structural analysis of AS genes in poplars”, we explored the factors with AS events. We revealed that the gene expression level was positively in relation to the AS events. See Fig 2 D.
Authors should consider adding a supplementary figure about the representation of different alternative splicing event with examples. I suggest a tool name “SpliceV” is easy to work on and give good resolution figures.
Reply: It was added
For analyzing gene ontology and pathway, authors can use different tools and compare the results.
Reply: The different tool has the specific implication in gene function annotation. Right here we used the regular GO and KEGG, which can significantly show the genes roles in plant resistance to high temperature.
Reviewer 2 Report
In this manuscript, the authors describe the results of studying episodes of alternative splicing (AS) in Populus tomentosa in response to high temperature.
This result was consistent with the physiological responses of poplar plants to heat stress, displaying that AS events in RNA degradation pathway was not worthy in abiotic resistance. It most likely gives us a fresh perspective on how stress resistance works and gives us some ideas for molecular breeding program techniques.
Research aimed at solving these problems is relevant.
As a comment, it should be noted that Figure 2 shows not the correlations that appear in its name, but the dependences of traits on the number of AS genes. Correlations have values from -1 to +1.
Author Response
Reviewer 2
As a comment, it should be noted that Figure 2 shows not the correlations that appear in its name, but the dependences of traits on the number of AS genes. Correlations have values from -1 to +1.
Reply: The description was modified
Reviewer 3 Report
The manuscript titled “Genome-wide analysis of alternative splicing events respond-ing to high temperature in Populus tomentosa “ is devoted to study of alternative splicing in Populus tomentosa TC1521's in response to short (24 h) exposition of high temperature. According to sequence characterization, alternative splicing was negatively correlated with GC content but favorably correlated with intron length, exon number, exon length, and gene transcription level, but statistical significance of such differences was not revealed. Compared to treatment at 25°C, 2001 distinct alternative splicing genes were discovered at 40°C. They were primarily enriched in the RNA degradation pathway and the valine, leucine and isoleucine degradation route, according to GO and KEGG enrichment analysis. There is no explanation why other high temperatures (30, 35°C) did not affected.
According to sequence characterization, alternative splicing was negatively correlated with GC content but favorably correlated with intron length, exon number, exon length, and gene transcription level, but statistical significance of such differences was not revealed.
Compared to treatment at 25°C, 2001 distinct AS genes were discovered at 40°C. They were primarily enriched in the RNA degradation pathway and the valine, leucine and isoleucine degradation route, according to GO and KEGG enrichment analysis. These findings demonstrated how the alternative splicing process might be severely impacted by high temperatures.
Despite the importance of reported data this manuscript needs significant improvement.
1) At Page 1, “For instances, OsHSFA2dI and OsHSFA2dII, for instance, are two splicing isoforms of the rice heat shock transcription factor OsHSFA2d.” – please remove the repeat “for instance(s)”. It is not good to start Introduction from particular cases of alternative splicing events without detailed explanation of the phenomena, and statistical significance of alternative splicing, explained at Page 2:
“AS is heavily involved in all phases of plant development. More than 63% of genes with multiple exons underwent AS procedure, according to an examination of 28 soybean samples taken at different developmental phases using RNA-sequencing technology [9]. An analysis of the bamboo shoot transcripts revealed that 60.74% of the genes were affected by AS events, with winter bamboo samples undergoing the most AS processes com-pared to samples from early, late, and mature development periods.”
Introduction is very important for the manuscript and should not jump to particular cases without good explanation of the history and theory of the subject.
2) At the part 2.2. Physiological index test “The poplar plants in highth of 50 cm were treated at 25℃, 30℃, 35℃, 40℃ for 24 h, respectively. The mature poplar leaves were sampled and used for the physiological index measurements. Three biological replicates were set for each treatment” – Please, explain, why these temperatures were selected, why only 24 h treatment was chosen, and when the tests and RNA isolation was done in relation to the treatment.
3) Page 11, 5. Conclusions “ Alternative splicing (AS) number and the AS pattern are significantly in response to high temperature in Populus tomentosa. These AS genes primarily enriched in the RNA degradation pathway and the valine, leucine and isoleucine degradation route, indicating AS quitely involved in regulation of plant resistance to abitic stress. Also, AS genes are quitely characterized in sequence such as GC content, intron length, exon number, exon length, and gene transcription level, which provide the point for the further AS mechanism exploration.”
- The conclusion that this study “provide the point for the further AS mechanism exploration” is very sad for such large and careful work. Please, re-write it and find some more important results from the analysis of obtained data.
Moderate editing of English language required.
Author Response
Reviewer3
According to sequence characterization, alternative splicing was negatively correlated with GC content but favorably correlated with intron length, exon number, exon length, and gene transcription level, but statistical significance of such differences was not revealed.
Reply: We did the correlation analysis via multiple comparison among the different AS gene groups. We described that in the method section.
Compared to treatment at 25°C, 2001 distinct alternative splicing genes were discovered at 40°C. They were primarily enriched in the RNA degradation pathway and the valine, leucine and isoleucine degradation route, according to GO and KEGG enrichment analysis. There is no explanation why other high temperatures (30, 35°C) did not affected.
Reply: According to our plant physiological test under temperature, populus was significantly affected at 40°C but less at 30°C and 35°C. We then majorly analyzed the AS gene functions at 40°C. We also displayed some new AS events occurred at 30°C and 35°C with the different amount and pattern, which were shown in section 3.5 and Fig.7.
According to sequence characterization, alternative splicing was negatively correlated with GC content but favorably correlated with intron length, exon number, exon length, and gene transcription level, but statistical significance of such differences was not revealed.
Reply: The multiple comparison among the different AS gene groups was conducted for each character. The different letters on the columns indicate the statistical significance at level P=0.05. Pls see Fig.2. The only GC content showed no significant difference between low and high AS genes but they are all significantly different with the non-AS genes.
Compared to treatment at 25°C, 2001 distinct AS genes were discovered at 40°C. They were primarily enriched in the RNA degradation pathway and the valine, leucine and isoleucine degradation route, according to GO and KEGG enrichment analysis. These findings demonstrated how the alternative splicing process might be severely impacted by high temperatures. Despite the importance of reported data this manuscript needs significant improvement
Reply: We did the careful improvement for our manuscripts as followings.
1) At Page 1, “For instances, OsHSFA2dI and OsHSFA2dII, for instance, are two splicing isoforms of the rice heat shock transcription factor OsHSFA2d.” – please remove the repeat “for instance(s)”. It is not good to start Introduction from particular cases of alternative splicing events without detailed explanation of the phenomena, and statistical significance of alternative splicing, explained at Page 2:
“AS is heavily involved in all phases of plant development. More than 63% of genes with multiple exons underwent AS procedure, according to an examination of 28 soybean samples taken at different developmental phases using RNA-sequencing technology [9]. An analysis of the bamboo shoot transcripts revealed that 60.74% of the genes were affected by AS events, with winter bamboo samples undergoing the most AS processes com-pared to samples from early, late, and mature development periods.” Introduction is very important for the manuscript and should not jump to particular cases without good explanation of the history and theory of the subject.
Reply: We modified the introduction section.
2) At the part 2.2. Physiological index test “The poplar plants in highth of 50 cm were treated at 25℃, 30℃, 35℃, 40℃ for 24 h, respectively. The mature poplar leaves were sampled and used for the physiological index measurements. Three biological replicates were set for each treatment” – Please, explain, why these temperatures were selected, why only 24 h treatment was chosen, and when the tests and RNA isolation was done in relation to the treatment.
Reply: The temperatures and treatments hours were based on our preliminary test when the leaf started wilting at 40℃. The total RNA isolation was done at same time. We modified the description in the method section.
3) Page 11, 5. Conclusions “ Alternative splicing (AS) number and the AS pattern are significantly in response to high temperature in Populus tomentosa. These AS genes primarily enriched in the RNA degradation pathway and the valine, leucine and isoleucine degradation route, indicating AS quitely involved in regulation of plant resistance to abitic stress. Also, AS genes are quitely characterized in sequence such as GC content, intron length, exon number, exon length, and gene transcription level, which provide the point for the further AS mechanism exploration.” - The conclusion that this study “provide the point for the further AS mechanism exploration” is very sad for such large and careful work. Please, re-write it and find some more important results from the analysis of obtained data.
Reply: The conclusion part was modified.